# Use of Technology to Aid Clinical Audit in an Asian Emergency Medical Services Department

**DOI:** 10.3390/healthcare9050491

**Published:** 2021-04-22

**Authors:** Qin Xiang Ng, Wesley Lok Kin Yeung, Joey Ai Meng Tay, Shalini Arulanandam

**Affiliations:** 1Emergency Medical Services Department, Singapore Civil Defence Force, 91 Ubi Ave 4, Singapore 408827, Singapore; tay_ai_meng@scdf.gov.sg (J.A.M.T.); shalini_arulanandam@scdf.gov.sg (S.A.); 2Department of Medicine, National University Hospital, 5 Lower Kent Ridge Rd, Singapore 119074, Singapore; wesley_yeung@nuhs.edu.sg

**Keywords:** emergency medical services (EMS), paramedicine, clinical audit, technology, proof-of-concept

## Abstract

Although clinical audit is generally accepted to be an essential part of quality review and continuous quality improvement, there are limited reports on and several barriers to the implementation of effective clinical audit in an emergency medicine services (EMS) organization. The barriers include the significant amount of time, resources, and effort often required to conduct the audit. In this paper, we present a technology-enabled clinical audit tool, termed Medical Service Transformation and Innovation Compass (MYSTIC), which has transformed the way the clinical audit is performed in our EMS department. MYSTIC is a Python program we developed in-house, that extracts data from data fields found in routine ambulance case records maintained by our paramedics, and automatically assigns “pass” or “fail” flags based on pre-defined audit criteria. Compared to previous manual auditing, implementation of the MYSTIC computerized audit system increased the coverage of cases undergoing audit from 10% to 100% of all EMS-attended cases, and we were able to promptly identify and address some deficits in training and knowledge amongst our paramedics.

## 1. Introduction

Clinical audit is thought to be a vital part of quality review and continuous quality improvement for healthcare provision [1,2]. After adopting a set of well-defined standards, the aim of the audit is to uncover gaps between actual practice and expected standards and thereby plan, design, and implement changes to improve the quality of care in wherever it is delivered.

Despite its potential value, there are several barriers to effective clinical audit, including the significant amount of time, resources, and effort often required to conduct an audit [3]. The operational nature of an EMS provider poses even greater challenges. Lack of protected time and organizational impediments have frequently been cited as barriers to audit [4], while facilitating factors which promote its success include the use of modern technology and robust case record systems [5] and improved links between routine data collection and audit [6]. 

In this proof-of-concept report, we present a technology-enabled clinical audit tool, named Medical Service Transformation and Innovation Compass (MYSTIC), which has transformed the way the clinical audit is conducted in our Emergency Medical Services (EMS) department. It is hoped that this will improve the organizational standards and delivery of prehospital emergency care.

## 2. Organizational Context

The Singapore Civil Defence Force (SCDF) is the national EMS organization in Singapore, which currently consists of a fleet of 84 ambulances and responds to more than 190,000 calls (to the national emergency ‘995’ hotline) every year. As a standing operating procedure, all EMS-attended cases are recorded by the paramedics on hardcopy ambulance case records, then transcribed into electronic forms (with the assistance of a digital pen) within 48 h and uploaded for audit and data analysis. Hardcopy records are filed for reference. Prior to November 2018, clinical audit of our EMS involved a manual, laborious audit of randomly selected cases and subsequent follow-up actions by a considerably small team of dedicated auditors. Due to the inherent complexity of the paramedic protocols and large call volume, only about 10% of total cases were audited annually. 

EMS utilization is expected to continue to increase as the population grows and ages [7]; therefore, it is imperative to harness technology to automate the audit process. This increases the efficiency and accuracy of the clinical audit, and more quickly identifies individual errors and systemic areas for improvement in clinical management. 

## 3. Proposed Solution

MYSTIC is a Python program developed in-house by our EMS Department that extracts data from data fields found in routine ambulance case records, as maintained by our paramedics, and assigns “pass” or “fail” flags based on pre-defined audit criteria. For example, at least two attempts to insert an intravenous (IV) cannula may be considered one of the “pass” criterions for the cardiac arrest protocol, while another “pass” criteria for the same protocol may be the successful insertion of an advanced airway device, e.g., oropharyngeal (OPA) or laryngeal mask airway (LMA). Cases that “fail” the computerized audit would then be flagged out and auditors could perform a more thorough cross-checking audit or deep dive into the data. 

With this computerized audit tool, the auditors would also be able to access the original ambulance case records and make corrections for documentation error, record justifications (if any) for criteria which have not met the “pass” standards, and note the final audit finding. Paramedics may be called up for audit interviews, with the interview outcome and remedial action recorded in the audit database, in accordance with the previously approved clinical audit framework.

As mentioned before, each criterion consists of two components: (1) a clinical criterion, which subsets the types of cases for which the criteria are applicable; and (2) an audit criterion, which dictates the action or standard to be met. This is illustrated in Figure 1. 

### Pilot Study

To test the feasibility of MYSTIC, a pilot deployment was conducted using MYSTIC to analyze 100% of EMS-attended cases from March to May 2020. A sample of the case checklist in MYSTIC is shown below (Figure 2). 

In Figure 2, we can see that when no LMA was inserted prior to transport of an out-of-cardiac hospital arrest (OHCA), it would be automatically flagged as “failed”. Additionally, in this case, the field “IV insertion” was mistakenly left out when the paramedics inputted data into MYSTIC and was thus automatically flagged up by the program. Having inputted “5” attempts in the following field, this prompted the auditors to revisit the original ambulance case record, and it was found that IV access was secured, and this was a data error, as specified in the “Result Justified?” field. Depending on the audit findings, we would also provide corresponding feedback to the paramedics on areas for improvement.

After referencing international literature [8,9] and consulting with emergency medicine specialists, we chose to focus on the following four critical interventions for our pilot: (1) attempted IV for patients with systolic blood pressure less than 90 mmHg; (2) attempted adjunct airway for patients with reduced consciousness (Glasgow coma scale (GCS) six or less); (3) provided supplemental oxygen for patients with oxygen saturation (SpO_2_) less than 94%; and (4) provided a bag-valve mask (BVM) for patients with respiratory rate (RR) of fewer than eight breaths per minute. 

In terms of the exclusion criteria, cases that were dead on arrival and those that refused or did not require conveyance were excluded from the pilot.

## 4. Results and Discussion

With the use of MYSTIC, we reported the following statistics for eligible EMS-attended cases from March through May, 2020 (*n* = 41,537). All figures were cross-checked by trained auditors. 

As seen in Figure 3, clinical results from MYSTIC could be automatically displayed in customized dashboards and aggregated over time to show trends and provide additional insights for individualized feedback or data-driven decisions. 

The auditors were briefly interviewed separately for qualitative feedback on the pilot. By computerizing the audit process, all seven of our auditors were of the consensus that MYSTIC greatly increased the audit coverage of cases, reduced the time taken for each manual audit, and enhanced the efficiency of the clinical audit cycle. Objectively, compared to the previous manual audit, implementation of the MYSTIC system enabled a 100% audit coverage of our cases. Moreover, due to prior scanning by MYSTIC, there was also more time for auditors to conduct a detailed audit of flagged or highlighted cases. 

Overall, MYSTIC enabled the identification of common areas of weakness and lapses across the board, so that these weaker areas could be addressed comprehensively in our continuous education and training sessions or through the refining or clarification of clinical protocols. From our audit of the four critical interventions, some examples of the problems identified and how they were addressed are discussed below.

First, we uncovered failure to insert oropharyngeal airway (OPA) in 8–11% of patients with reduced consciousness (GCS six or lower). It is important to maintain an open airway in patients with a reduced level of consciousness, as recommended in our training protocols. When delved into, reasons for this omission included crew misinterpreting a patient’s baseline low conscious level and poor pre-morbid state as exclusion criteria. This misconception and the need to secure the airway in patients with reduced consciousness was promptly addressed in our training sessions.

Secondly, we found delays in recognizing hypotension in some cases, which led to the inadvertent omission of IV insertion. Paramedics with consistently poor IV success rates were also identified to receive extra hands-on training. Through this deliberate and systematic approach, we achieved some small improvements over time, as evidenced by a slight reduction in the number of flagged cases using the same clinical criteria from March, April, and May, 2020 (5.6%, 4.0% and 2.6% of total cases, respectively). However, admittedly, the difference in the overall number of “pass” cases was not statistically significant when comparing March and May, 2020 (*p* = 0.457).

This is only a pilot study; further planned intervention is clearly necessary. We also lack further empirical data, and our findings remain preliminary and unadjusted for potential confounders. More objective data are required, and it remains to be seen if our targeted audit and interventions truly improve clinical outcomes. MYSTIC relies on data extracted from discrete fields keyed in by the paramedics; therefore, documentation error is another potential limitation, and we are also unable to ascertain documentation accuracy at this time. In our pilot, we had 5–10% missing or erroneous data. Another challenge for MYSTIC is to keep focus on identifying systemic areas for the training and refining of protocols, rather than coming across as punitive or overly penalize individuals based on their audit performance alone. A myopic emphasis on clinical audit criteria may drive some individuals to manipulate the system by false documentation to “pass’” or avoid audit, which ultimately could not be detected by our audit methods. To counter this, we plan to refresh or rotate our audit criteria every three months.

Moving forward, we have also expanded our audit criteria to include 15 important interventions for specific approaches and EMS protocols. They are outlined in Table 1.

## 5. Conclusions

Based on these pilot results, SCDF’s executive leadership supported operationalizing and scaling MYSTIC. With growing EMS utilization and the increasing complexity of paramedicine protocols, it is important and necessary to harness technology to automate the clinical audit process. The audit process ensures that certain clinical standards are continually achieved by our paramedics, as well as identifies knowledge and practice gaps. This should theoretically help focus and drive training and quality improvement initiatives. In the longer term, we will work towards digitizing all ambulance records and developing an integrated clinical case management framework incorporating real-time, seamless documentation by our paramedics and an intelligent audit system.

## Figures and Tables

**Figure 1 healthcare-09-00491-f001:**
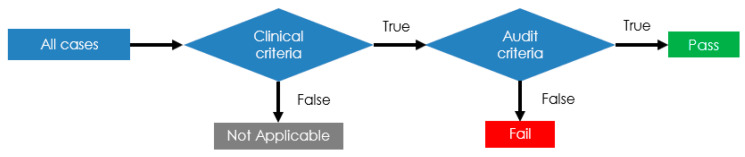
MYSTIC audit logic.

**Figure 2 healthcare-09-00491-f002:**
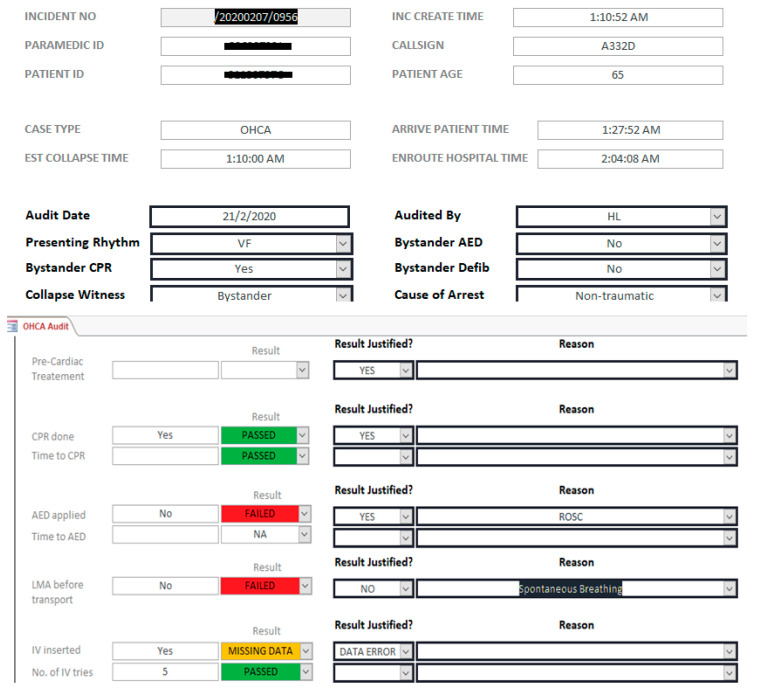
Clinical case audit for out-of-cardiac hospital arrest (OHCA) cases in MYSTIC.

**Figure 3 healthcare-09-00491-f003:**
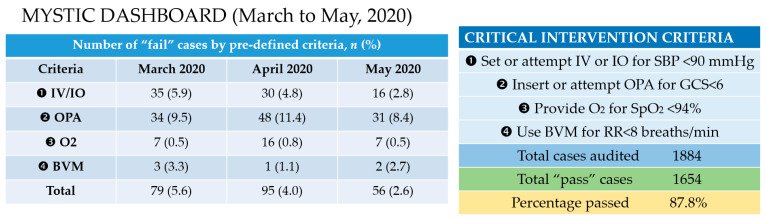
Results of MYSTIC audit from March through May 2020.

**Table 1 healthcare-09-00491-t001:** Updated MYSTIC criteria selection record.

No.	Protocol	Criteria
1.	critical_Circulation	Standby activated when SBP is less than 90 mmHg
2.	critical_Circulation	ECG done/attempted when heart rate is less than 50 bpm
3.	critical_Circulation	Standby activated if there is bradycardia/signs of shock
4.	protocol_ACS	GTN administered/attempted if SBP is more than 90 mmHg
5.	protocol_ACS	Aspirin administered/attempted
6.	protocol_ACS	ECG done/attempted
7.	protocol_CCF	ECG done/attempted
8.	protocol_CCF	GTN administered/attempted if SBP is more than 90 mmHg
9.	protocol_diabetic	Dextrose/oral glucose administered/attempted when capillary blood glucose is less than 4 mmol/L
10.	protocol_diabetic	IV normal saline administered/attempted when capillary blood glucose is more than 16 mmol/L
11.	protocol_stroke	Physical examination (CPSS) documented
12.	protocol_stroke	Standby activated when CPSS is positive
13.	protocol_stroke	Capillary blood glucose checked
14.	protocol_AMS	Capillary blood glucose checked
15.	protocol_AMS	ECG performed/attempted

Abbreviations: ACS, acute coronary syndrome; AMS, altered mental state; bpm, beats per minute; CCF, congestive cardiac failure; CPSS, Cincinnati Prehospital Stroke Scale; ECG, electrocardiogram; GTN, Glyceryl trinitrate; SBP, systolic blood pressure.

## Data Availability

The datasets generated during and/or analyzed during the current study are available from the corresponding author on reasonable request.

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
