# Peer review of "Use of Technology to Aid Clinical Audit in an Asian Emergency Medical Services Department"

_healthcare, 2021, doi:10.3390/healthcare9050491_

Round 1

Reviewer 1 Report

Please pay attention to the syntax, spelling and punctuation. Please analyze the text in this respect. Besides, the part relating to the literature analysis, and therefore the results of other scientific research, requires a substantial increase. The research results should be preceded by an analysis of the scientific achievements of other authors. It is worth noting that issues of clinical audit studies are extensively presented in the literature. 

The author wrote: "By computerizing the audit process, the auditors were of the consensus that ..." Such a task causes dissatisfaction in the reader and prompts to ask at least such questions as How many auditors, where, when, how was the audit conducted? Were these interviews, questionnaires, document examinations?

The author wrote: "As this is only a pilot concept paper, we lack further empirical data, and our findings remain preliminary and unadjusted for potential confounders." It is worth noting that such pilot concept articles should follow certain universal requirements concerning the presentation of research results. By now, the article looks like a description of certain assumptions of the system, but since it is presented, it must be presented in such a form that it is understandable for the readers, not only the authors or reviewers.

The author wrote: "A myopic emphasis on clinical audit criteria may drive some individuals to 'game' the system by false documentation to" pass' "or avoid audit, which ultimately could not be detected by our audit methods". Such a statement generates other research questions. From the reader's perspective, this question arises: what to do in such a situation? What preventive measures should be introduced?

The author wrote: "Compared to previous manual auditing, implementation of the MYSTIC computerized audit system increased .. (....) and we were able to promptly identify and address some deficits in training and knowledge amongst our paramedics." Where are the exact results of the research that allow the readers to get acquainted with the system's benefits, such as, as the author points out, the quick identification of the competency gap among employees?

Author Response

Comment 1: Please pay attention to the syntax, spelling and punctuation. Please analyze the text in this respect. Besides, the part relating to the literature analysis, and therefore the results of other scientific research, requires a substantial increase. The research results should be preceded by an analysis of the scientific achievements of other authors. It is worth noting that issues of clinical audit studies are extensively presented in the literature.

Reply 1: Thank you for the comments. We have done a close edit of the entire manuscript for syntax, spelling and punctuation. We have also expanded on our introduction paragraph by noting the issues of clinical audit highlighted by previous literature. “Despite its potential value, there are several barriers to effective clinical audit, including the significant amount of time, resources and effort often required to conduct the audit [3]. Lack of protected time and organisational impediments have frequently been cited as barriers to audit [4], while facilitating factors which promote its success include the use of modern technology and case record systems [5] and improved links between routine data collection and audit [6].”

Comment 2: The author wrote: “By computerizing the audit process, the auditors were of the consensus that ...” Such a task causes dissatisfaction in the reader and prompts to ask at least such questions as How many auditors, where, when, how was the audit conducted? Were these interviews, questionnaires, document examinations?

Reply 2: Thank you for the comments. We apologise for the ambiguity. We have now clarified that, “The auditors were interviewed separately for qualitative feedback on the pilot. By computerizing the audit process, all seven of our auditors were of the consen-sus that MYSTIC greatly increased the audit coverage of cases, reduced the time taken for each manual audit and enhanced the efficiency of clinical audit cycle.”

Comment 3: The author wrote: "As this is only a pilot concept paper, we lack further empirical data, and our findings remain preliminary and unadjusted for potential confounders." It is worth noting that such pilot concept articles should follow certain universal requirements concerning the presentation of research results. By now, the article looks like a description of certain assumptions of the system, but since it is presented, it must be presented in such a form that it is understandable for the readers, not only the authors or reviewers.

Reply 3: Thank you for the comments. We fully agree with the reviewer that the article must be understandable for readers. The objective of the article was to illustrate the feasibility of automating the audit process. We have restructured our article as a concept paper, which more accurately reflects the pilot nature of our study and our stated objective of demonstrating the feasibility of an automated audit process. We hope this is understandable for the wider audience.

Comment 4: The author wrote: "A myopic emphasis on clinical audit criteria may drive some individuals to 'game' the system by false documentation to" pass' "or avoid audit, which ultimately could not be detected by our audit methods". Such a statement generates other research questions. From the reader's perspective, this question arises: what to do in such a situation? What preventive measures should be introduced?

Reply 4: Thank you for the comments. We have added that, “To counter this, we plan to refresh or rotate our audit criteria every 3 months.”

Comment 5: The author wrote: "Compared to previous manual auditing, implementation of the MYSTIC computerized audit system increased .. (....) and we were able to promptly identify and address some deficits in training and knowledge amongst our paramedics." Where are the exact results of the research that allow the readers to get acquainted with the system's benefits, such as, as the author points out, the quick identification of the competency gap among employees?

Reply 5: Thank you for the comments. We hope to clarify that the MYSTIC tool proposed in this concept paper merely automates the process of audit and analyses cases based on the predetermined audit criteria. This then allows our auditors to do a further deep dive into the “fail” cases to identify the gaps. For example, “we uncovered failure to insert oropharyngeal airway (OPA) in 8 to 11% of patients with reduced consciousness (GCS 6 or lower). It is important to maintain an open airway in patients with reduced level of consciousness, as recommended in our training protocols. When delved into, reasons for this omission included crew misinterpreting a patient’s baseline low conscious level and poor pre-morbid state as an exclusion criteria. This misconception and the need to secure a patent airway in patients with reduced consciousness was promptly addressed in our training sessions.” However, we also acknowledge that “the difference in the overall number of “pass” cases was not statistically significant when comparing March and May, 2020 (p=0.457).” This was likely due to the fact that further planned intervention is required.

Reviewer 2 Report

This is a manuscript about the use of technology for quality review and continuous improvement. The system implemented is called MYSTIC (medical service transformation and Innovation compass) developed by the home institution of the authors. This system was then used for auditing EMS protocols. The records pulled data fields from ambulance case records and assigned a pass/fail based on audit predefined criteria. 

  • attempted IV for patients with systolic blood pressure less than 90 mmHg, 
  • attempted adjunct airway for patients with reduced consciousness (Glasgow coma scale  (GCS) 6 or less)
  •  provided supplemental oxygen for patients with oxygen saturation (SpO2) less than 94%, and 
  • provided bag-valve mask (BVM) for patients with respiratory rate (RR) less than 8 breaths per minute.

Major Feedback

  • It is unclear if the data pulled from the EMS database is manually entered for screening or if the data fields were automatically created. The authors should describe how the extraction occurs and if there are errors during this process. What is the sensitivity and specificity of extraction fields in MYSTIC from EMS records? An expanded methodology section may help clarify this question
  • 102: The authors state that the time taken for audits decreased, could the authors quantify this ? 
  • The mystic dashboard in page 4 should be a figure with p values that describe differences if any. It seems as if some of these pre-defined criteria did not decrease and the authors should comment on why that may be the case (for example, O2, or why OPA insertion fails increased in APril 2020). 
  • The authors would benefit the reader by providing utilization rates of MYSTIC over time by case reviewers, and help illustrate how non mystic case review occurs. How did this software change the review process?

Minor Feedback

  • The manuscript is well written overall
  • Figure 2 is unclear and could use a more clear screenshot of the software. The black highlighting is hard to read
  • Figure 2: It seems as if “Result justified” has a “DATA ERROR” field. Can the authors explain why? Is this related to the MISSING DATA field in REsult? If so, how often did this occur?

Author Response

Comment 1: It is unclear if the data pulled from the EMS database is manually entered for screening or if the data fields were automatically created. The authors should describe how the extraction occurs and if there are errors during this process. What is the sensitivity and specificity of extraction fields in MYSTIC from EMS records? An expanded methodology section may help clarify this question.

Reply 1: Thank you for the comment. We have clarified in the study limitations that, “MYSTIC relies on data extracted from discrete fields keyed in by the paramedics, documentation error is another potential limitation and we are also unable to ascertain documentation accuracy at this time.”

Comment 2: The authors state that the time taken for audits decreased, could the authors quantify this?

Reply 2: Thank you for the comment. We apologize for the ambiguity. We are unable to quantify this as the auditors were only interviewed separately for qualitative feedback on the pilot as stated in the manuscript.

Comment 3: The mystic dashboard in page 4 should be a figure with p values that describe differences if any. It seems as if some of these pre-defined criteria did not decrease and the authors should comment on why that may be the case (for example, O2, or why OPA insertion fails increased in April 2020).

Reply 3: Thank you for the comment. Figure 2 was taken directly from our programme to show the dashboard function, which aggregates the “pass” rate for the individual audit criteria and overall “pass” rate for each month. It does not have any additional statistical testing function at present. However, we acknowledge in the text that the difference between the months was not statistically significant and have now added this in our results and discussion section, “However, the difference in the overall number of “pass” cases was not statistically significant when comparing March and May, 2020 (p=0.457).” This was likely due to the fact that further planned intervention is required.

Comment 4: The authors would benefit the reader by providing utilization rates of MYSTIC over time by case reviewers, and help illustrate how non mystic case review occurs. How did this software change the review process?

Reply 4: Thank you for the comment. MYSTIC had 100% utilization rate by the case reviewers in this pilot study and is already implemented throughout our EMS for all case reviewers.

Comment 5: Figure 2 is unclear and could use a clearer screenshot of the software. The black highlighting is hard to read.

Reply 5: Thank you for the comment. We have enhanced the quality of the image.

Comment 6: It seems as if “Result justified” has a “DATA ERROR” field. Can the authors explain why? Is this related to the MISSING DATA field in Result? If so, how often did this occur?

Reply 6: Thank you for the comment. We apologise for the ambiguity. We have now clarified that, “In Figure 2, we can see that when no LMA was inserted prior to transport of an out-of-cardiac hospital arrest (OHCA), it would be automatically flagged as “failed”. In this case also, the field “IV insertion” was mistakenly left out when the paramedics inputted into MYSTIC, and thus automatically flagged up by the programme. Having inputted 5 attempts in the following field, this prompted the auditors to revisit the original ambulance case record and it was found that IV access was secured and this was a data error as specified in the “Result Justified?” field.”

We also added that, “In our pilot, we had 5 to 10% of missing or erroneous data.”

Reviewer 3 Report

As authors reported clinical audit is essential to identify deficiencies in current practice and pursue improvement in outcomes.The manuscript has demonstrated the automated audit is more effective compared to a manual audit. 

The results demonstrated a significant increase in the number of cases audited electronically and ability to identify deficiencies in clinical skills. However the study did not demonstrate how the audit helped in improving patient outcomes and did not report any comparative data from prior manual audits.

But overall this study confirms the advantage of an automated audit to review extensive medical records compared to a manual process and may provide more accurate data and methods to improve patient outcomes.

Author Response

Thank you for the comments. We appreciate you reviewing our manuscript.

Round 2

Reviewer 1 Report

Dear Author

On behalf of the editorial team, I want to thank you for submitting your research to the Journal. I truly appreciate having an opportunity to read your work and to become aware of a topic that interests you.
You should first understand that the bar for publication in this Journal is set high. It is critical that a manuscript increase our theoretical understanding – or at a minimum – clearly identify and articulate critical gaps and potential contributions to the scientific literature. It must extend what we know in a meaningful way. 
Unfortunately, my final review of your paper suggests the article is not well-written from the scientic point of view.  Thus, in my opinion, your manuscript fails to meet the minimum standards referenced above. I wish to respect your valuable time waiting for reviewers to respond with a negative outcome.  As a result, it is my decision to bring the review process to a close.  I think the Journal is not able to publish your paper, nor are we able to request that you revise and resubmit the manuscript to us for a next-round review. 

I realize that this news will come as a disappointment to you, especially in light of the amount of work that you have expended to complete this manuscript. Nonetheless, I believe that the decision conveyed to you is appropriate. 

This manuscript is a resubmission of an earlier submission. The following is a list of the peer review reports and author responses from that submission.

Round 1

Reviewer 1 Report

I propose to make numerous stylistic and punctuation changes. For example, the noun phrase clinical audit seems to be missing a determiner before it. Consider adding an article. 

You wrote: "Although clinical audit is generally accepted to be an essential part of quality review and continuous quality improvement, there are limited reports on and several barriers to the implementation of effective clinical audit in an emergency medicine services (EMS) organization. The barriers include the significant amount of time, resources and effort often required to conduct the audit. In this paper, we present a technology-enabled clinical audit tool, termed Medical Service Transformation and Innovation Compass (MYSTIC), which has transformed the way the clinical audit is performed in our EMS department. MYSTIC is a Python programme we developed in-house, that extracts data from data fields found in routine ambulance case records maintained by our paramedics, and automatically assigns “pass” or “fail” flags based on pre-defined audit criteria. Compared to previous manual auditing, implementation of the MYSITC computerized audit system increased the percentage of cases undergoing audit from 10% to 100% of all EMS-attended cases, and we were able to promptly identify and address some deficits in training and knowledge amongst our paramedics". 

Consider rephrasing: "Although the clinical audit is generally accepted as an essential part of quality review and continuous quality improvement, there are limited reports on several barriers to implementing effective clinical audit in an emergency medical services (EMS) organization. The barriers include the significant amount of time, resources and effort often required to conduct the audit. This paper presents a technology-enabled clinical audit tool, termed Medical Service Transformation and Innovation Compass (MYSTIC), which has transformed the way the clinical audit is performed in our EMS department. MYSTIC is a Python programme we developed in-house that extracts data from data fields found in routine ambulance case records maintained by our paramedics and automatically assigns "pass" or "fail" flags based on pre-defined audit criteria. Compared to previous manual auditing, the MYSITC computerized audit system's implementation increased the percentage of cases undergoing audit from 10% to 100% of all EMS-attended cases. We were able to promptly identify and address some deficits in training and knowledge amongst our paramedics."

Next example. You wrote: "It is widely acknowledged that clinical audit is a vital part of quality review and continuous quality improvement for healthcare provision [1,2]." Your sentence may be hard to follow. Consider rephrasing. "It is widely acknowledged that clinical audit is vital for quality review and continuous quality improvement for healthcare provision [1,2]."

For example: "the aim of the audit is". Consider rephrasing. 

The authors present a proposal to publish a scientific article. It means that the methodology should be significantly expanded. The authors wrote: "After referencing international literature [5] and consulting ...." It seems that literature studies should be significantly expanded. The analysis of one piece of research work seems to be not enough. 

"The audit process ensures certain clinical standards are continually achieved by our paramedics, as well as identify certain knowledge and practice gaps." The audit process seems to help achieve the stated objectives by identifying areas of weakness. It also seems that in order to formulate a conclusion about the "knowledge gaps", it is necessary to thoroughly rely on the results of the research conducted so far (literature study) and first identify the research gap. Maybe the authors intended to write that audit enables identifying an information gap or a knowledge management gap. The article is interesting. However, it needs significant improvement. The theoretical part should be significantly expanded, which will better understand the practical part of the article. What is an audit? Can it use a concept of weak signals? These are just some questions that arise from the analysis of this article. Then the article will gain significant cognitive value. I propose to analyze the article in light of stylistic and punctuation requirements. Besides, Please, specify the research problems that have been solved due to the research and presented in the article.

Reviewer 2 Report

Ng et al. present a technology-enabled clinical audit tool, termed Medical Service Transformation and Innovation Compass (MYSTIC) for EMS department. • The paper is technically sound but exactly study type, inc & excl criteria, sensitivity and specificity calculation is missing • The methods are appropriate and properly conducted but require the name of the tests/algorithm required for calculating the results. • Results: Line 89: “all EMS-attended cases”: how many? • The statistical analysis of the data not clear • The manuscript could be revised to address any potential limitations • Discussion: Limitations are missing

Reviewer 3 Report

Article that presents an acceptable introduction, material and methods are not clearly described and are not good, a low "N" for the number of cases attended, as indicated in the article despite only showing data from two months. The results do not respond emphatically to the stated objective. Poorly structured article and they are excessively short, due to the little amount of data that the article provides.